# Wildebeest-Derived Malignant Catarrhal Fever: A Bovine Peripheral T Cell Lymphoma Caused by Cross-Species Transmission of *Alcelaphine Gammaherpesvirus 1*

**DOI:** 10.3390/v15020526

**Published:** 2023-02-13

**Authors:** Meijiao Gong, Françoise Myster, Willem van Campe, Stefan Roels, Laurent Mostin, Thierry van den Berg, Alain Vanderplasschen, Benjamin G. Dewals

**Affiliations:** 1Laboratory of Immunology-Vaccinology, Faculty of Veterinary Medicine, FARAH, ULiège, Avenue de Cureghem 10, B-4000 Liège, Belgium; 2Laboratory of Parasitology, Faculty of Veterinary Medicine, FARAH, ULiège, Avenue de Cureghem 10, B-4000 Liège, Belgium; 3Sciensano, Scientific Directorate Infectious Diseases in Animals, Experimental Center Machelen, Kerklaan 68, B-1830 Machelen, Belgium

**Keywords:** gammaherpesvirus, malignant catarrhal fever, lymphoproliferative disease, bovine, CD8^+^ T lymphocytes, latency

## Abstract

Gammaherpesviruses (γHVs) include viruses that can induce lymphoproliferative diseases and tumors. These viruses can persist in the long term in the absence of any pathological manifestation in their natural host. *Alcelaphine gammaherpesvirus 1* (AlHV-1) belongs to the genus *Macavirus* and asymptomatically infects its natural host, the wildebeest (*Connochaetes* spp.). However, when transmitted to several susceptible species belonging to the order *Artiodactyla*, AlHV-1 is responsible for the induction of a lethal lymphoproliferative disease, named wildebeest-derived malignant catarrhal fever (WD-MCF). Understanding the pathogenic mechanisms responsible for the induction of WD-MCF is important to better control the risks of transmission and disease development in susceptible species. The aim of this review is to synthesize the current knowledge on WD-MCF with a particular focus on the mechanisms by which AlHV-1 induces the disease. We discuss the potential mechanisms of pathogenesis from viral entry into the host to the maintenance of viral genomes in infected CD8^+^ T lymphocytes, and we present current hypotheses to explain how AlHV-1 infection induces a peripheral T cell lymphoma-like disease.

## 1. Wildebeest-Derived Malignant Catarrhal Fever, a Fatal Lymphoproliferative Disease

Wildebeest-derived malignant catarrhal fever (WD-MCF) is caused by cross-species transmission of *Alcelaphine gammaherpesvirus 1* (AlHV-1), a virus species that infects the two species of wildebeest: blue wildebeest (brindled gnu, *Connochaetes taurinus*) and black wildebeest (white-tailed gnu, *C. gnou*) [1,2,3]. The geographical distribution of the blue wildebeest extends virtually throughout Sub-Saharan Africa as far as the Orange River. The greatest densities of blue wildebeest populations are found in the savannah of East Africa, where around 1.5 million animals travel nearly 3000 km each year during periods of great migration [4,5]. Black wildebeests inhabit a more restricted geographical area in southern Africa and currently only exist in captivity in nature reserves and game parks [6].

AlHV-1 naturally infects wildebeests and early reports in East Africa demonstrated that the prevalence of infection in free-ranging wildebeests approached 100% [7,8,9]. Importantly, AlHV-1 infection in wildebeests is totally asymptomatic. Interestingly, a recent small-scale study in zoological collections of wildebeests in France reported a seroprevalence of 46% [10], suggesting that AlHV-1 circulates in wildebeests in captivity. AlHV-1 can be transmitted from infected wildebeests to other susceptible ruminant species, including cattle. In those species, AlHV-1 is responsible for WD-MCF, a deadly lymphoproliferative disease. AlHV-1-induced WD-MCF is indistinguishable from sheep-associated MCF caused by *Ovine gammaherpesvirus 2* (OvHV-2) from both clinical and pathological point of views. Synonyms of MCF include the Afrikaans name *snotsiekte*, but also *malignant catarrh* or *gangrenous coryza* [11].

**Figure 1 viruses-15-00526-f001:**
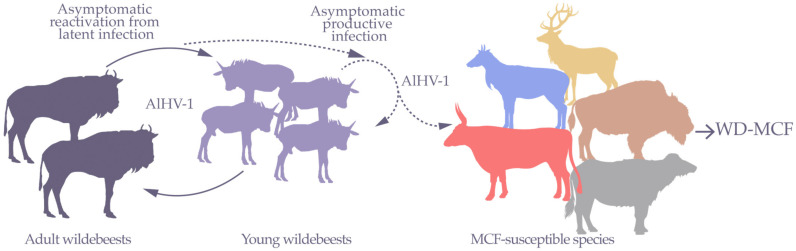
Epidemiological relationships of AlHV-1 transmission between wildebeests and species susceptible to WD-MCF. Wildebeests are asymptomatic carriers of AlHV-1. Following reactivation events, AlHV-1 is mainly transmitted horizontally to wildebeest calves. In young wildebeests, AlHV-1 replicates and is mainly transmitted by nasal secretions. Young wildebeests are infectious with peak infectivity at around 4 months before latent infection is established for the rest of their life. During these periods of viral shedding, AlHV-1 can be transmitted to several MCF-susceptible species, including cattle, bison, deer, buffalo, and other related antelopes in which AlHV-1 infection is responsible for the development of WD-MCF (in free-ranging or captive conditions). Susceptible animals are epidemiological dead ends for disease transmission. Adapted from a figure created in Biorender.com.

From an epidemiological point of view, young wildebeests seem to become infected before the fourth month of life, mainly by the nasal route from animals excreting free viral particles [12]. During this period, wildebeest calves are usually infectious (Figure 1). Indeed, free viral particles have been observed in nasal secretions up to this age before decreasing upon the appearance of neutralizing antibodies [13,14], as also observed during OvHV-2 infection in sheep flocks [15]. The virus subsequently persists in a latent form, probably within certain types of blood mononuclear cells. Although no study has been performed to identify cell tropism of latent infection in wildebeests, it is possible that AlHV-1 also infects T lymphocytes, as has been observed in OvHV-2-infected sheep [16]. In adults, experimental infections suggested that different types of stress, such as confinement, diet modification, or corticosteroids, can lead to the excretion of viral particles in nasal secretions [17,18]. Vertical transmission is rare but cannot be completely excluded [2,6,19]. Strikingly, while infection is widespread in wildebeests, no clinical sign has ever been reported upon infection and they do not develop WD-MCF.

Oral and nasal secretions of young wildebeests contain AlHV-1 viral particles and are likely the main source of transmission to susceptible species (Figure 1) [13,14]. WD-MCF-susceptible species include several phylogenetically related species. Data regarding AlHV-1 seroprevalence in WD-MCF-susceptible species is not readily available. Thus, it remains unclear whether infection inevitably leads to WD-MCF clinical manifestations in these species. However, it is accepted that when WD-MCF clinical signs develop, mortality will follow in most cases. In East Africa, WD-MCF cases generally occur during the dry season, a period associated with the peak of the wildebeest birth period [20]. Although seroprevalence and/or viral detection of MCF-associated viruses (MCFV) have been reported in approximately 33 different species of ruminants of the families *Bovidae*, *Cervidae*, *Giraffidae* and *Antilocapridae*, independent of association to disease or not [21], it is currently not possible to differentiate the causal MCFV species due to cross-reactive antibodies. Interestingly, ruminant species developing MCF all belong to the subfamilies *Bovinae* and *Antilopinae* or are members of the family *Cervidae* (Figure 2). There is currently no data to explain why only members of these phylogenetically related species reportedly develop clinical disease upon infection, but it is possible that genetically conserved traits would explain the susceptibility or resistance to developing WD-MCF [22]. In addition to ruminants, some laboratory animals have been used to reproduce WD-MCF clinical signs [3]. Among them, rabbits represent the most reliable experimental model since intravenous or intranasal inoculation results in typical WD-MCF clinical signs and lesions after 3 to 4 weeks on average, depending on the inoculation dose [23,24,25,26]. Importantly, WD-MCF-susceptible animals represent an epidemiological dead end and cannot transmit the virus, likely due to the absence of free viral particles in the lesions [3,27].

**Figure 2 viruses-15-00526-f002:**
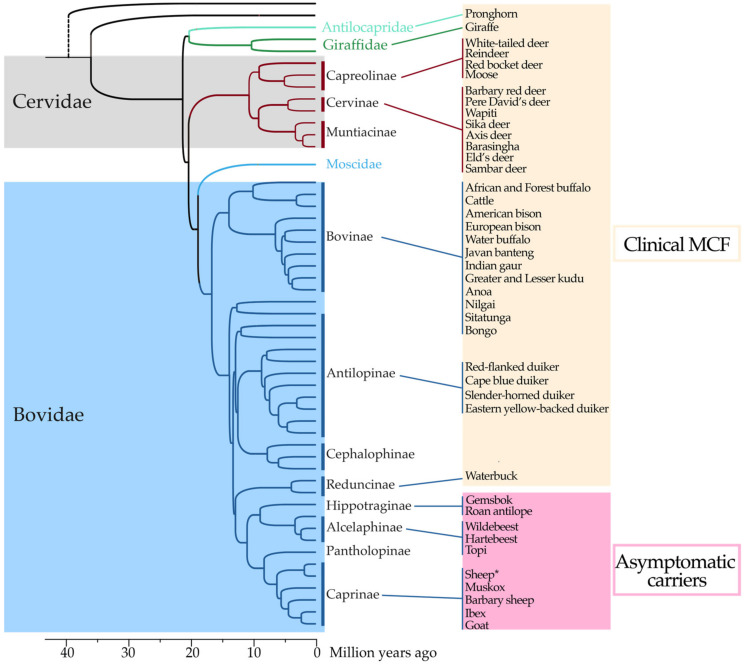
Phylogenetic relationships of ruminant species that develop MCF clinical signs and lesions or asymptomatically carry MCFV. The majority of MCF-susceptible species belong to the family *Cervidae* or subfamilies *Bovinae* and *Antilopinae* (family *Bovidae*). Strikingly, asymptomatic carriers responsible for MCFV transmission belong to closely related subfamilies *Hippotraginae*, *Alcelaphinae,* and *Caprinae* (family *Bovidae*) [15,28,29,30,31,32,33,34,35]. *Reported cases exist of sheep developing clinical signs of MCF [36,37,38]. Phylogenetic tree of ruminants adapted from ref. [39].

**Figure 3 viruses-15-00526-f003:**
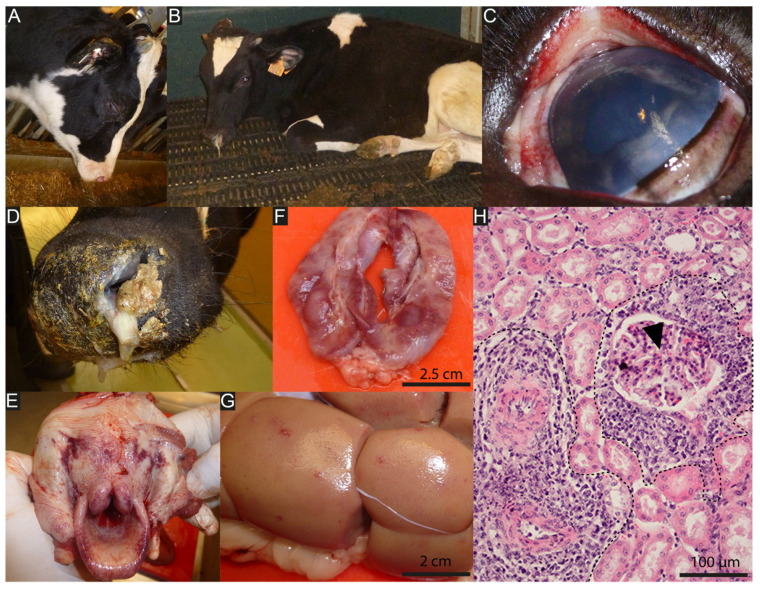
Typical lesions of bovine WD-MCF induced by AlHV-1 infection. (**A**,**B**) Calves showing typical head and eye form of WD-MCF, with apathy, ptyalism, mucopurulent discharge, and muzzle crusting. (**C**) Corneal opacity and mucosal petechia. (**D**) Detail of the muzzle illustrating the presence of adherent fibrino-necrotic material that partially occludes the nasal orifices. (**E**) Ulcerative and necrotic pharyngitis and laryngitis. (**F**) Longitudinal section through an inguinal lymph node. Note the loss of normal tissue structure of the organ and necrotic appearance. (**G**) Irregularly distributed white to reddish foci of a few millimeters in diameter deform the surface of the kidney. (**H**) Hematoxylin and eosin-stained kidney section highlighting the accumulation of lymphoblastic cells around small arterioles (dashed curves). Arrowhead indicates renal corpuscle.

While various forms of MCF have been described, the clinical and pathological manifestations of both WD-MCF and sheep-associated MCF are similar. The World Organization for Animal Health (WOAH) reports MCF clinical signs as variable, which can be acute to chronic [40]. In 1930, Götze described the main clinical forms of MCF while studying the sheep-associated form of the disease in Europe [3]. The main patterns of disease manifestations are peracute, head and eye, intestinal, cutaneous, and neurological. Clinical signs only develop after incubation periods that can vary from 7 to 46 days and sometimes longer [3,41,42,43]. The disease induced by AlHV-1 in cattle is overwhelmingly a head and eye form (80% of cases) [3,44]. In contrast, some susceptible deer species and bison will develop a peracute form upon infection with either AlHV-1 or OvHV-2. Once clinical signs start, cattle will die after around a week or more and deer or bison will die within 2 to 3 days [11]. The typical head and eye form is characterized by muco-purulent ocular and nasal discharge, congestion and crusting of the muzzle, and erosion of the nasal, oral, and ocular mucosa (Figure 3). The peracute form is rather characterized by either the absence of clinical signs or generalized depression followed by diarrhea 12 to 24 h prior to death [31]. The disease is almost always fatal, either due to death induced by the infection or decision to euthanize due to intense suffering of the sick animals. After experimental infection, the duration of illness between the onset of fever and death was estimated at 12 ± 4 days after intravenous inoculation of heparinized blood [44]. In general, the severity of MCF clinical signs will increase with the duration of survival. A generalized adenomegaly is observed, with inguinal and axillary lymph nodes becoming hypertrophic and palpable. The adenomegaly is generally associated with persistent hyperthermia (>40 °C), anorexia, and dyspnea. Progressive bilateral corneal opacity, starting at the periphery, is often observed [3]. Moreover, skin ulceration on the udder can be observed, as well as nervous signs such as hyperesthesia, incoordination nystagmus, and head pressing.

In general, leukopenia is observed just before or at the time of hyperthermia and is accompanied by relative lymphocytosis associated with a decrease in the number of small lymphocytes and marked proportional increase in the number of large lymphoblastic cells [3]. At autopsy, generalized adenomegaly is observed, particularly marked in the region of the head and neck. Petechial hemorrhages on the tongue and buccal mucosa as well as in the gastrointestinal and respiratory tracts can be noticed. Circular white to darker foci can be macroscopically observed at the renal surface and localized in the cortex upon incision. The lymph nodes are firm and may show hemorrhagic and/or necrotic foci after incision. The spleen is enlarged with a prominence of white pulp [3]. At the histological level, the lesions are characterized by generalized subacute peri-vasculitis, marked by the accumulation of large lymphoblastic cells in the perivascular spaces of small arterioles and venules in both lymphatic and non-lymphatic tissues such as the liver, kidney, lung, heart, or brain, which are sometimes associated with necrotic lesions [3,45,46,47] (Figure 3). Phenotypic characterization of the infiltrating lymphoblastic cells in the tissue revealed that most of the accumulating cells in the perivascular spaces express T cell markers such as CD2, CD3, and CD8 [25,41,45,48,49,50,51], which is associated with the observed expansion of activated CD8^+^ T lymphocytes in the peripheral blood that precedes MCF clinical signals after AlHV-1 infection in cattle [41] and also in the rabbit experimental model [24,25,26,27,41,48]. Thus, in terms of pathological observations, MCF resembles a peripheral T cell lymphoma [45].

## 2. AlHV-1 Is a *Gammaherpesvirinae*, a Subfamily Responsible for Lymphoproliferative Diseases

Viruses belonging to the order of *Herpesvirales* have been classified into three families: *Mallacoherpesviridae*, comprising a species infecting a mollusk; *Alloherpesviridae*, including viruses of fish and amphibians; and *Herpesviridae*, members of which infect various species of birds, reptiles, and mammals [52,53]. One of the important characteristics of herpesviruses is their ability to persist in their respective natural hosts long term. Indeed, in addition to their ability to replicate in permissive cells and ensure viral multiplication, herpesviruses can establish a latent infection during which the viral genome is maintained in the cell nucleus as an episome, in the absence of production of infectious viral particles [54].

The *Herpesviridae* family is divided into three subfamilies: *alpha-, beta-*, and *gammaherpesvirinae*. Unlike alpha- and beta-herpesviruses that establish their latent cycle mainly within non-dividing or poorly dividing cells, gammaherpesviruses (γHVs) establish their latent cycle within actively dividing cells, such as B or T lymphocytes [54]. Based on genomic and biological characteristics, γHVs have been divided into four genera: *Lymphocryptovirus*, *Rhadinovirus*, *Percavirus*, and *Macavirus* (Table 1) [52]. Within the genus *Lymphocryptovirus*, Epstein-Barr virus (EBV or *Human gammaherpesvirus 4*, HHV-4) infects more than 90% of the human population [55]. EBV is the causative agent of infectious mononucleosis and can induce neoplastic manifestations such as Hodgkin’s or Burkitt’s lymphomas and nasopharyngeal carcinoma. The genus *Rhadinovirus* contains three main viral species associated with tumor development. First, herpesvirus saimiri (HVS or *Saimiriine gammaherpesvirus 2*, SaHV-2) is considered the prototype of its kind [56]. SaHV-2 causes a fatal disease associated with the proliferation of CD4^+^ T lymphocytes in certain primates [56]. Although humans are not susceptible to the disease, SaHV-2 can transform T lymphocytes from different animal species in vitro, including those from humans. Kaposi’s sarcoma-associated herpesvirus (KSHV or *Human gammaherpesvirus 8*, HHV-8) is a virus of considerable importance to humans. KSHV infection has a prevalence of around 30% in Sub-Saharan Africa and Mediterranean countries [57,58]. Initially identified within Kaposi’s sarcoma lesions, KSHV is also responsible for B cell neoplastic manifestations such as primary effusion lymphoma and multicentric Castleman’s disease [59,60]. Human γHVs are strongly restricted to their natural hosts, which makes it difficult to study human γHV infection and lymphomagenesis. *Murid gammaherpesvirus 4* (MuHV-4) has been extensively used as an experimental model in laboratory mice and has largely contributed to the understanding of several key mechanisms used by γHVs to infect and persist in their host [61]. Nonetheless, studies on MuHV-4 infection have not unraveled all facets of human γHV infection and the development of experimental models using humanized mice to study EBV or KSHV infection are already providing important information [62].

**Figure 4 viruses-15-00526-f004:**
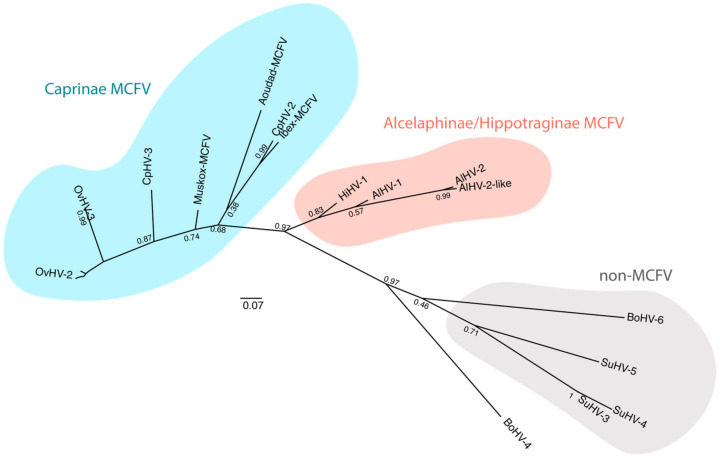
Phylogenetic analysis of members of the genus *Macavirus*. Phylogenetic tree based on the predicted ORF 9 (DNA polymerase) DNA sequence of known members of the genus *Macavirus* [63]. Sequences were aligned with Clustal W and the phylogenetic tree was generated by using the Maximum Likelihood and Tamura-Nei model [64]. The tree with the highest log likelihood (−1785.80) is shown. The tree is drawn to scale, with branch lengths measured in the number of substitutions per site. This analysis involved 33 nucleotide sequences. There were 156 positions in the final dataset. Evolutionary analyses were conducted using MEGA11 [65] and the final tree was constructed using FigTree 1.4.4. AlHV-1, KX905134 obtained from an ox; AlHV-2, KF274499 isolated from a Topi antelope (*Damaliscus korrigum*); AlHV-2-like, AY092762 obtained from a hartebeest (*Alcelaphus buselaphus*); HiHV-1, DQ083950 obtained from a roan antelope (*Hippotragus equinus*); CpHV-2, AF275941 and CpHV-3, AF181468 obtained from goats; Ibex-WD-MCFV, AY212112 obtained from an Ibex; Aoudad-WD-MCFV, DQ083952 obtained from a Barbary sheep; Muskox-WD-MCFV, AY212111 obtained from a muskox; OvHV-2, AY839756, DQ198083, HQ450395, HM216472, HM216465, HM216458, AF327831, AF031812, HM216470, HM216464, HM216460, HM216456, HM216455, EU309722, EU078708, HM216461, HM216468, and AF181468 obtained from sheep; OvHV-3, MN068215 and MN068216 obtained from bighorn sheep; BoHV-6, AF327830 obtained from cattle; SuHV-3, AF478169, SuHV-4, AY170317 and SuHV-5, AY170315 obtained from pigs; BoHV-4, NC_002665, *Rhadinovirus* obtained from cattle. BoHV-4 was used as an outgroup to root the tree. The bar shows the frequency of substitutions per site.

The *Percavirus* and *Macavirus* genera were created in 2009 and include viruses formerly classified among *Rhadinovirus* species or newly identified species [52]. While the genus *Percavirus* includes viruses infecting equids and carnivores, the genus *Macavirus* includes viruses associated with WD-MCF (*malignant catarrhal fever virus*), such as AlHV-1 and OvHV-2, as well as phylogenetically related viruses. All viral species of the genus *Macavirus* that are associated with MCF share a conserved epitope (15-A), which is present in envelope glycoprotein B (gB) [66,67,68]. Phylogenetic analysis of known members of the genus *Macavirus* generates three main groups (Figure 4).

One group of viruses is not associated with MCF, including *Bovine gammaherpesvirus 6* and *Suid gammaherpesvirus 3, 4,* and *5*. Another group is composed of MCFV of ruminants, from the subfamily *Caprinae*. A last group contains MCFV of African antelopes, from the subfamilies *Alcelaphinae* or *Hippotraginae*, including AlHV-1.

## 3. Unravelling WD-MCF Pathogenesis from AlHV-1 Genomic Sequence

In 1960, Plowright et al. isolated the causative agent of wildebeest-derived WD-MCF, which was later referred to as AlHV-1 [2]. At that time, structural and cytopathological characteristics led to its classification as a herpesvirus [69]. AlHV-1 has the characteristic structure of herpesviruses, with a nucleocapsid of icosahedral symmetry of approximately 118 nm in diameter surrounding a double-stranded DNA molecule and covered with a tegument and envelope containing viral glycoproteins. The total diameter of virions ranges from 140 to 220 nm [70,71]. A preliminary characterization of the genome classified AlHV-1 within the genus *Rhadinovirus* [72]. In 1997, the entire genome sequence of strain C500 was released, and its genome was annotated by homology with the genome of SaHV-2 [73]. Strain C500 was isolated from an ox developing WD-MCF [74]. AlHV-1 now belongs to the genus *Macavirus* since its creation in 2009 by the International Committee on Taxonomy of Viruses (ICTV) [52].

AlHV-1 has a genome of approximately 150,000 bp, consisting of a 130,608 bp long unique region with low GC content (low-DNA region, L-DNA) flanked by several repeat units (±12–15 repeats) of approximately 1,108 bp with high GC content (high-DNA region, H-DNA) [73,75]. Another strain, named WC11, was initially isolated from a young wildebeest [72,76]. Strain WC11 was extensively passaged (>1000 times) in cell culture, which led to its attenuation and inability to induce MCF. Interestingly, the genome of strain WC11 was recently sequenced and genomic analysis revealed a nucleotide sequence almost identical to that of strain C500 (>99% sequence homology) [26,77], with a few deletions and mutations that could explain its attenuation. The L-DNA region of the AlHV-1 genome has a lower observed/expected CpG ratio than that of the H-DNA, indicating higher methylation of this region despite the abundance of promoter regions. AlHV-1 L-DNA potentially comprises 73 open reading frames (ORFs) [26,73], which are distributed over the entire length of the genome except for two regions of non-repeated sequences that do not seem to code for any ORF. The organization is co-linear with other *Rhadinovirus* members and 60 ORFs are homologous to SaHV-2 genes that were organized in 5 blocks (Figure 5) [78]. Blocks I, II, and IV contain ORFs that are conserved across the sequenced species of the family *Herpesviridae* and are involved in the lytic cycle and replication. Blocks III and V comprise ORFs that are conserved only in γHVs.

### 3.1. γHV Latency Is an Essential Mechanism for WD-MCF Pathogenesis

Viral genes regulating the lytic cycle and latency establishment are conserved in γHVs and have been studied in the context of AlHV-1 infection. ORF57 is located within block IV and is therefore conserved among all herpesviruses. Expressed very early during the lytic cycle of γHVs, it regulates other viral genes at a post-transcriptional level. AlHV-1 ORF57 encodes an immediate-early protein and its ability to regulate ORF50 promoter activity has been demonstrated [79]. ORF50 is positioned in block III and encodes the reactivation transactivator (Rta) protein. In γHVs, Rta is an immediate-early viral gene that promotes the transactivation of various genes, including immediate-early, early, and late genes. This protein is also the only known essential protein for conversion to the lytic cycle in *Rhadinovirus* species [80].

**Figure 5 viruses-15-00526-f005:**
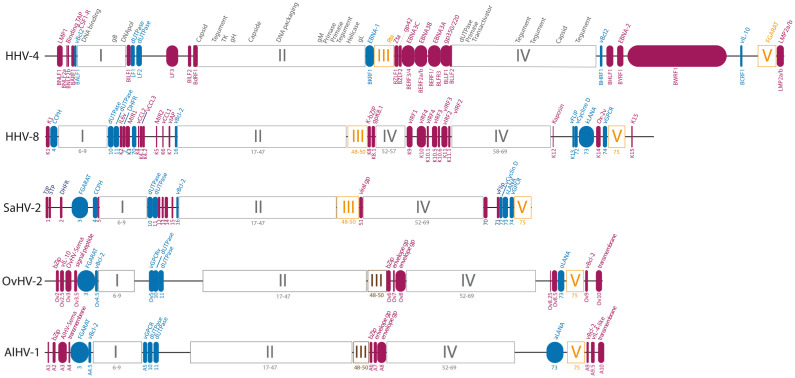
Genomic organization of the coding region of several selected γHVs. The genome of AlHV-1 strain C500 (NC_002531, KX905135) [73,75] has been aligned with the genomes of OvHV-2 (NC_007646) [81], SaHV-2 (NC_001350) [82], KSHV (NC_009333) [83], and EBV (NC_007605) [84]. Blocks conserved within herpesviruses are shown in grey and those conserved among γHVs are shown in orange. Within the variable zones, the genes shared by several γHVs are in blue and the genes specific to each viral species are in magenta. The different ORFs are indicated below the representation of the genome and the corresponding proteins above. Viral genes homologous to cellular genes are indicated by their abbreviation preceded by the letter “v”. DNApol: DNA polymerase; Bcl-2: B cell lymphoma 2, anti-apoptotic protein; CCL: CC chemokine; CCPH: homolog of complement control proteins; CSF1-R: colony stimulating factor receptor 1; DHFR: dihydrofolate reductase; EBNA: Epstein-Barr virus nuclear antigen; FGARAT: ribotid formylglycinamide amido-transferase; Flip: FLICE inhibitor protein, anti-apoptotic protein; gp: glycoprotein; GPCR: G-protein coupled receptor; IAP: apoptosis inhibitor protein; IL-6: interleukin 6; IL-10: interleukin 10; IRF: interferon regulatory factor; KbZip: KSHV “leucine zipper” basic domain protein; LANA: latency-associated nuclear antigen; LMP: latent membrane protein; MIR: immune recognition modulator; GMP: genomic maintenance protein; Rta: reactivation transactivator; Sema: semaphorin; Tip: SaHV-2 tyrosine kinase interacting protein; STP: SaHV-2 transformation associated protein; TAP: transporter associated with the presentation of major histocompatibility complex class I antigens; TK: tymidine kinase; Zta: ZEBRA transcription activator.

In AlHV-1, ORF50 also encodes an immediate-early protein, which has been shown to increase the activity of ORF6 and ORF57 gene promoters similarly to its homologs [85]. We can thus consider that AlHV-1 ORF50 promotes lytic infection, although the functions of ORF50 and ORF57 have not been studied in the context of viral infection [85]. To investigate the role of specific genomic regions, the entire AlHV-1 genome has been cloned as an infectious and pathogenic bacterial artificial chromosome (BAC) from pathogenic strain C500 [86]. Interestingly, full genome sequencing of the AlHV-1 BAC clone revealed a genomic duplication of ORF50 and the bZIP-protein-coding A6 gene in the H-DNA. A duplication of ORF50 and A6 is also present in the highly passaged strain WC11, suggesting probable selection of such duplicating events during replication in cell culture [75]. Moreover, ORF50 mRNA expression could only be detected in low abundance in bovine lymph nodes during WD-MCF [41], suggesting poor activation of the lytic cycle in vivo.

In addition to genes involved in viral reactivation, the establishment and maintenance of latent γHV infection is mediated by one specific gene that encodes genome maintenance protein (GMP). γHV GMPs include the EBNA-1 protein encoded by EBV BKRF1 and the latency-associated nuclear antigen (LANA)-1 encoded by KSHV ORF73 [87]. GMPs are essential for γHVs to maintain their genomes and persist in a latent state within dividing cells, such as lymphocytes. GMPs are involved in the initiation of episomal replication of viral genomes and are also determinant in tethering viral episomes to cellular chromosomes, thus ensuring the distribution of viral genomes within daughter cells during cell division. In AlHV-1, ORF73 encodes a large aLANA protein of 1300 amino acids that is expressed during WD-MCF [24,41]. Importantly, it was demonstrated that CD8^+^ T lymphocytes expand during WD-MCF and carry the viral genome as episomes. Moreover, deletion of the coding sequence of AlHV-1 ORF73 or insertion of nonsense mutations to impair aLANA expression rendered AlHV-1 unable to induce WD-MCF in the rabbit model [41], whereas its replication in cell culture was not altered. These important results demonstrated that the maintenance of AlHV-1 genomes via aLANA in vivo is an essential mechanism to induce WD-MCF, likely through the establishment of a latent-like infection in proliferating T lymphocytes.

In addition to their role in genome maintenance, GMPs evade immune recognition by self-regulating their own epitope presentation to the immune system [87]. The mechanisms involved vary according to the viral species but overall result in self-inhibition of the presentation of GMP epitopes via the MHC-I. Hence, this mechanism allows latently infected cells to evade detection by cytotoxic T lymphocytes [87,88,89,90,91]. AlHV-1 aLANA contains a *cis*-acting glycin/glutamate (GE)-rich repeat domain that regulates the protein synthesis of aLANA itself, thereby inhibiting the presentation of an antigenic peptide linked to it [92]. Interestingly, a mutated form of aLANA devoid of the GE-rich domain resulted in the effective peptide presentation in a cell-based in vitro model. However, the *cis*-acting immune evasion mechanism mediated by aLANA GE is dispensable for WD-MCF. Indeed, expression of an aLANA-ΔGE form during AlHV-1 infection in vivo did not affect WD-MCF induction in the rabbit model [92]. Thus, aLANA likely escapes immune recognition in latently infected cells in the natural host, but the absence of the aLANA *cis*-acting immune evasion mechanism is not sufficient to avoid disease development in WD-MCF-susceptible species.

### 3.2. Studying AlHV-1-Specific Genes and Non-Coding RNAs to Investigate WD-MCF Pathogenesis

In addition to genes being conserved in γHVs, some genes are either conserved only among some genera or specific to viral species. For instance, γHVs encode homologs of cellular genes that are involved in cell adhesion and cell tropism switch, immune evasion, or lymphomagenesis (Figure 5) [55,93]. The AlHV-1 genome encodes 12 predicted ORFs that are specific to AlHV-1, which were identified by the prefix “A”. Interestingly, these specific genes have positional and sequencial orthologs in the genome of OvHV-2 (Table 2).

#### 3.2.1. Regulation of Cell Tropism during Host Entry Is Essential for WD-MCF and Regulated by Accessory Envelope Glycoproteins A7 and A8

Herpesviruses enter target cells using complex mechanisms involving several essential or accessory envelope glycoproteins [54]. The functions of these glycoproteins range from tethering virions to target cells, to mediating envelope-membrane fusion, and finally, to delivering viral capsids into the cytoplasm. Virion entry has been extensively studied in EBV infection [94]. Five glycoproteins are involved in EBV entry into B cells, including the attachment glycoprotein gp350/220, the receptor binding protein gp42, and the core fusion proteins gH/gL and gB [95,96]. After binding of the attachment protein gp350/220 to CR2, the type II glycoprotein encoded by gp42 contains a C-type lectin-like domain that interacts with host HLA class II and activates gH/gL, which triggers fusion of the cellular lipid bilayer [97,98]. Interestingly, functional homologs of gp350/220 were also identified in other γHVs, including KSHV K8.1, MuHV-4 gp150, and *Bovine gammaherpesvirus 4* (BoHV-4) gp180 [99,100,101]. However, homologs of gp42 were only identified in *Lymphocryptovirus* members and *Macavirus* species. Indeed, AlHV-1 A7 gene is predicted to encode a C-type lectin-like protein [26]. In EBV, gp42 and gp350/220 mediate tropism switch from epithelial cells to B lymphocytes [96]. When analyzing the genome sequence of the attenuated strain WC11, it appeared that a region of 2174 bp containing A7 and a fraction of A8 genes was deleted [26], which could explain WC11 attenuation. Whereas A7 is likely a gp42 homolog, A8 is a positional homolog of the gene encoding gp350/gp220 in EBV and could also be involved in AlHV-1 tropism regulation. Indeed, impaired A7 expression in a recombinant AlHV-1 BAC clone resulted in increased abundance of cell-free virions in cultures, whereas impaired A8 expression resulted in a significant defect in cell-associated propagation of AlHV-1 in cell culture [26]. Moreover, OvHV-2 Ov8-encoded protein could enhance cell-membrane fusion by the gB/gH/gL entry complex [102]. In addition to these observations, the absence of A7 and/or A8 expression during AlHV-1 infection in the rabbit model resulted in complete attenuation of the virus, suggesting that viral entry in the host is regulated by these accessory envelope glycoproteins that are both essential in WD-MCF pathogenesis [26].

**Table 2 viruses-15-00526-t002:** AlHV-1- and other *Macavirus*-specific genes, encoded proteins, and known functions in WD-MCF.

AlHV-1	OvHV-2	% id. ^a^	BoHV-6	% id. ^a^	SuHV3	% id. ^a^	Protein ^b^	Description ^c^
A1	-	-					-	-
A2	Ov2	56	Bov2	30	-	-	bZIP protein	Presence of a “leucine zipper” domain, a possible transcription factor, non-essential for WD-MCF induction in rabbits [103]
-	Ov2.5	-	Bov2.5	-	-	-	viral IL-10	Stimulates mast cell proliferation and inhibits IL-8 production by macrophages [104]
A3	Ov3	50	-	-	-	-	viral sema7A	Secreted, binds to plexin C1 and inhibits dendritic cell migration, non-essential for WD-MCF induction in rabbits [105,106]
-	Ov3.5	-	-	-	-	-	Unknown	Signal peptide
A4	-	-	-	-	-	-	SaHV-2 STP homolog	Unknown
A4.5	Ov4.5	50	Bov4.5	35	-	-	viral Bcl-2	Putative anti-apoptotic function
A5	Ov5	49	Bov5	41	-	-	vGPCR	Inhibits CREB signaling pathway, non-essential for WD-MCF induction in rabbits [107]
A6	Ov6	28	Bov6	-	A6/BZLF1	33.9	bZIP protein (Zta homolog)	Presence of a “leucine zipper” domain, a possible transcription factor
A7	Ov7	60	Bov7	35	A7/BZLF2	32.0	EBV gp42 homolog	Virus glycoprotein, essential for WD-MCF induction in rabbits [26]
A8	Ov8	41	Bov8	25	A8/BLLF1	25.3	EBV gp350/220 homolog	Virus glycoprotein, essential for WD-MCF induction in rabbits [26]
-	Ov8.25	-	-	-	-	-	Targets mitochondria	Putative role in caspase-dependent apoptosis and necrosis [108]
A9	Ov9	50	Bov9	27	-	-	viral Bcl-2	Putative anti-apoptotic function
A9.5	-	-	-	-	-	-	viral IL-4-like	Encodes a secreted glycoprotein [109]
A10	Ov10	22	-	-	-	-	SaHV-2 Tip homolog	Putative oncogene

^a^ Information about the proteins that are encoded by specific genes and functions in WD-MCF. ^b^ Identification based on sequence homology or positional homology. bZIP: basic leucine zip; STP: STP oncoprotein of SaHV-2; IL-10: interleukin 10; sema7A: Semaphorin 7A; Bcl-2: B cell lymphoma 2, anti-apoptotic protein; GPCR: G-protein coupled receptor; gp: glycoprotein; gp42 and gp350/220: EBV envelope glycoproteins. IL-4: interleukin 4; Tip: Tip oncoprotein of SaHV-2. ^c^ Description of known function. Role in WD-MCF pathogenesis relates to studies using recombinant viruses generated with the AlHV-1 BAC clone.

#### 3.2.2. Secreted AlHV-1 Semaphorin Homolog Is Dispensable for WD-MCF Induction

Semaphorins represent a superfamily of membrane or secreted proteins involved in signal transmission, mainly in the nervous system [110,111]. Interestingly, several members of this family are expressed by immune cells and have been involved in various immune functions [112]. Among semaphorins engaged in the immune response, semaphorin 7A (sema7A; also known as CD108) is a glycosylphosphatidylinositol (GPI)-anchored signaling protein that appears to interact with α1β1 integrins expressed by monocytes and macrophages to drive T cell-mediated inflammatory responses [113]. Furthermore, sema7A has been associated with the inhibition of T-cell responses in studies using sema7A-deficient mice [113]. Interestingly, the sema7A sequence was initially identified based on its sequence similarity with a secreted viral semaphorin expressed by poxviruses and named A39R [114]. A39R can bind to the receptor plexin C1 and shares sequence similarities with AlHV-1 protein-coding gene A3. Importantly, besides poxviruses, members of the genus *Macavirus* are the only viruses known to encode a viral semaphorin [106]. The A3-encoded semaphorin, known as AlHV-sema, is a secreted protein expressed during viral infection and is able to bind plexin C1 to induce rearrangements of the cytoskeleton via a Rock-dependent cofilin phosphorylation mechanism [105]. Importantly, AlHV-sema can interact with dendritic cells and inhibit their migration from peripheral sites to the draining lymph node, suggesting an immune evasion mechanism to delay T-cell activation [105]. However, impaired expression of AlHV-sema during AlHV-1 infection in vivo did not affect the ability of the virus to induce WD-MCF [105], suggesting that AlHV-sema is likely involved during infection of AlHV-1 in natural hosts and not in WD-MCF pathogenesis.

#### 3.2.3. AlHV-1 Encodes Specific Protein-Coding Genes Potentially Involved in the Induction of Lymphoproliferation

G protein-coupled receptors (GPCRs) are transmembrane receptors able to transfer signals from the cell surface to the inside of the cell [115]. Viral GPCR homologs are also encoded by poxviruses and herpesviruses [116]. Previous studies have demonstrated that the cAMP response element-binding protein (CREB), activator protein 1 (AP-1), NF-KB, and NFAT transcriptional potential are modulated by the G protein-coupled receptors (GPCR) of EBV and KSHV [117,118]. Similarly, it has been demonstrated that the viral GPCR encoded by AlHV-1 A5 constitutively mediates CREB activation, even though A5 has been shown to be dispensable for WD-MCF [107].

AlHV-1 encodes the basic leucine zipper (bZIP) family of proteins that likely regulate host and/or virus gene transcription, leading to changes that could result in lymphoproliferation [119]. Interestingly, AlHV-1 A2 is a bZIP family protein [103]. Deletion of A2 from the AlHV-1 genome does not alter viral replication nor the ability of AlHV-1 to induce WD-MCF in a rabbit model, suggesting that it is not essential on its own in driving WD-MCF pathogenesis [103]. However, lymphoblastoid cell lines generated from peripheral mononuclear cells isolated from rabbits infected with A2-deleted viruses and cultured in the presence of IL-2 showed reduced expression of cytotoxicity markers [103]. These results suggested that A2 could contribute to the activation of infected T lymphocytes. In addition, the AlHV-1 A6 gene is a positional homolog of EBV Zta [75], a bZIP family protein that has been shown to regulate reactivation. A6 is duplicated in the genome of strains C500 and WC11 [75], and AlHV-1-infected bovine CD8^+^ T lymphocytes have high levels of A6 mRNA expression (unpublished). These data indicate that A6 is expressed during WD-MCF in vivo and could play a role in the regulation of gene expression during infection.

Several oncogenic viruses manipulate the host cell cycle and cell death signals to persist in infected cells in the long term. Inhibition of apoptosis is often described as a mechanism that promotes lymphomagenesis induced after EBV or KSHV infection [120]. Bcl-2 and Bcl-x L are the two members of the Bcl-2 family. The death-promoting homolog is the BCL-x L isoform, while the death-inhibiting homolog is the Bcl-2 isoform [121]. Several γHV species encode viral Bcl-2 (vBcl-2) protein-coding genes that lengthen the longevity of infected lymphocytes to complete the viral life cycle. In EBV infection, in addition to the BHRF1-encoded Bcl-2 homologue being produced to inhibit apoptosis, the BALF1-encoded Bcl-x L homologue is also produced during lytic infection to negatively regulate apoptosis [122,123,124]. Such mechanisms help to maintain viral homeostasis in the host and open the door for the induction of tumors [122,123,124]. Importantly, several additional viral genes, such as KSHV ORF16 and MuHV-4 M11 genes, share sequence similarities with cellular Bcl-2 [120]. Likewise, AlHV-1 A4.5 and A9 genes also encode Bcl-2 homolog proteins [11]. As in EBV, the co-expression of Bcl-2 homologs to manipulate apoptosis of infected cells might represent a key mechanism regulating the long-term persistence of AlHV-1 infection in vivo and could contribute to the fatal lymphoproliferation of infected T cells in WD-MCF-susceptible species.

In addition to the potential regulation of gene transcription by bZIP proteins and reprogramming of cell death by vBcl-2s, γHVs also encode oncogenes, resulting in the regulated expression of transmembrane proteins that interfere with the activation signals of B or T lymphocytes [125]. For example, EBV LMP2A is localized in lipid rafts and interferes with B cell signaling by interacting with the Src family kinase Lyn [125]. Similarly, KSHV K15 inhibits the NF-κB and mitogen-activated protein kinase (MAPK) pathways, resulting in the inhibition of the T-cell signaling pathway [125]. By interacting with the SH2 domain of the Src family, SaHV-2 STP protein triggers STAT3 phosphorylation, whereas Tip protein is phosphorylated by binding to the SH3 structural domain of Lck and subsequently activates the NFAT, NF-KB, and STAT-1 and -3 pathways [126,127,128,129]. Critically, SaHV-2 recombinant viruses unable to express Tip fail to induce T-cell lymphoproliferation upon infection in vivo and in vitro [129], but the exact mechanism remains unknown since mutation of the Tip SH3 domain does not affect tumor development in vivo [126]. AlHV-1 A4 and A10 genes, respectively, encode putative type I or type II membrane proteins of unknown function [73]. Both A4 and A10 have sequence characteristics that are reminiscent of EBV, KSHV, and SaHV-2 oncogenic proteins (LMP1, K1, and STP share similarities with A4; LMP2A, K15, and Tip share similarities with A10) [125]. Although A4 sequence homology is poor with its positional homologs, the A10 gene is predicted to encode a 50 kDa type II transmembrane protein with a long cytoplasmic domain containing multiple tyrosine residues, which is a conserved feature in LMP2A, K15, and Tip oncoproteins [125]. In addition, protein prediction analyses of A10 identified an immunoreceptor tyrosine-based activation motif (ITAM) and a Src homology 3 domain (SH3) motif in its cytoplasmic domain, which makes it an interesting candidate viral gene that could interfere with T0cell signaling during AlHV-1 infection.

#### 3.2.4. AlHV-1 Encodes microRNAs Expressed in Lymphoblastoid Cell Lines

In the non-coding region of the AlHV-1 genome located downstream of ORF11, a large cluster of microRNAs (miRNAs) was identified by genome sequencing analysis [130]. The presence of a cluster of miRNAs was also predicted at the same position in OvHV-2 and *Equid gammaherpesvirus 2* (EHV-2). As observed for other viral miRNAs, the sequences of AlHV-1, OvHV-2, or EHV-2 miRNAs are not conserved [131]. The expression of viral miRNAs has notably been confirmed during infection with EBV and KSHV [131,132]. MiRNAs are mostly expressed during latency. In addition to their action on cellular transcripts, miRNAs could also target certain viral transcripts and regulate the maintenance of latency [133]. Interestingly, the expression of 8 miRNAs was detected in bovine lymphoblastoid cells cultured in vitro from animals infected with OvHV-2 and developing WD-MCF [134]. Similarly, in bovine lymphoblastoid cells established from AlHV-1-infected calves, deep sequencing of small RNAs identified 32 potential miRNAs. These miRNAs were mainly encoded on the reverse strand of the genome and were primarily distributed in two main clusters [135]. One cluster included four miRNAs at the left end of the genome and the second cluster encoded 28 miRNAs in the region devoid of protein-coding genes between ORF11 and ORF17 [135]. Despite their number and the reported roles of viral miRNAs in lymphomagenesis after Marek disease virus infection [131], the deletion from the AlHV-1 genome of the 28 microRNAs encoded in the second cluster did not affect WD-MCF pathogenicity in the rabbit model, suggesting that miRNAs contribute to the regulation of viral infection but are not essential in the pathogenesis of WD-MCF [135].

## 4. AlHV-1 Establishes a Latency-Like Infection in CD8^+^ T Lymphocytes to Induce Their Activation and Proliferation during WD-MCF

The data obtained from experimental infection in the rabbit model using a luciferase-expressing AlHV-1 strain indicated that respiratory infection is followed by multiplication at the primary site before establishment of latent infection that lasts during the incubation period [27,41]. Viremia is strictly associated with mononuclear cells, which likely originates from lymphoid tissues since high viral DNA loads can be detected in these tissues [70,136]. Viremia is detectable from 9 to 17 days after inoculation and, on average, 7 days before the onset of pyrexia. DNA viral copy numbers rise sharply from the onset of clinical signs until the death of the sick animal [24,27,41].

The understanding of WD-MCF pathogenesis has long been limited to the description of histological lesions and analysis of lymphocyte populations isolated and propagated from the tissues of animals developing WD-MCF [137]. Although lymphoblastoid cell lines can be derived from purified mononuclear cells of an WD-MCF-developing bovine for several days, their long-term maintenance in culture is facilitated by the addition of IL-2. The generated lymphoblastoid cell lines are mainly composed of large T lymphocytes referred to as “large granular lymphocytes” (LGLs) [135]. LGLs are infected with AlHV-1 and possess cytotoxic activity that does not appear to be limited by MHC-I-dependent restriction and does not require antigenic stimulation [136,137]. Although it is interesting to note that LGLs constitutively express the Src-like kinases Fyn and Lck, the fact that LGLs are maintained in culture in the presence of IL-2 makes it difficult to distinguish the effects due to AlHV-1 infection from the IL-2-mediated differentiation of early described lymphokine-activated killer (LAK) cells [138]. Thus, phenotypic description of LGLs must be taken with caution and future work should better investigate the phenotypic changes of T lymphocytes during WD-MCF in cattle. Nonetheless, and despite the difficulty of maintaining LGLs in culture long term, recent data have established that AlHV-1 targets CD8^+^ T lymphocytes and induces WD-MCF by promoting the activation and uncontrolled proliferation of infected cytotoxic T lymphocytes, which results in peri-vascular infiltration and associated lesions [24,27,41].

The expansion of CD8^+^ T lymphocytes is therefore at the center of WD-MCF pathogenesis. Indeed, in vivo incorporation of 5-bromo-2’-deoxyuridine demonstrated severe CD8^+^ T-lymphocyte proliferation in the peripheral blood, spleen, and lymph nodes after AlHV-1 infection in a rabbit model [24]. Consistent with analyses performed on tissues from WD-MCF cases in cattle [51], Anderson et al. also observed an infiltration of predominantly CD8^+^ T lymphocytes in the lymphatic and non-lymphatic organs of the rabbit model [139]. In addition, a detailed analysis of the phenotype of these proliferating CD8^+^ T lymphocytes during WD-MCF in rabbits highlighted their activated phenotype, with increased expression of activation markers, perforin, and interferon γ [25], observations that were also confirmed after experimental infection in cattle [41,140]. However, the precise role of cytotoxicity in the induction of the lesions observed during WD-MCF remains to be determined, as well as the mechanisms induced by AlHV-1 that trigger T-cell lymphoproliferation resulting in a peripheral T cell lymphoma-like disease.

Initially, the main hypothesis to explain the lesions developed during WD-MCF was based on the very low detection of viral infection using DNA in situ hybridization [76] and by immunostaining for viral antigens [141] (10^−4^ to 10^−6^ infected cells, respectively). According to this hypothesis, a low number of infected cells would lead to the dysregulation of surrounding uninfected lymphocytes. The lesions would therefore be caused by uninfected cells. More recently, several key studies have reinforced another theory proposing that the lesions are rather due to the proliferation of infected lymphocytes, such as in lymphomas caused by other γHVs. According to this model, a latent-like infection would be directly responsible for the dysregulation, activation, and proliferation of the infected cells themselves by unknown mechanisms. Therefore, the observed WD-MCF lesions would result from the expansion and infiltration of infected cells.

The rabbit experimental model has largely contributed to identifying the tropism of AlHV-1 as being mainly restricted to CD8^+^ T lymphocytes, in which episomal genomes of AlHV-1 are maintained. Episomal maintenance of γHV genomes in proliferating lymphocytes is a characteristic of latency-based lymphomagenesis [24,41,55]. AlHV-1 genomes were detected in an episomal conformation in cattle during WD-MCF [41], supporting the hypothesis of a latent-like infection. In addition, AlHV-1 aLANA protein expression could be detected in most T cells expanding in lymph nodes during WD-MCF in calves or rabbits, an observation that was also confirmed by detection of ORF73 transcripts using single-cell RT-PCR from isolated CD8^+^ T lymphocytes [41]. Moreover, the use of a recombinant virus expressing luciferase as a transgene demonstrated that the viral infection could be detected in all organs analyzed in the rabbit model, and the bioluminescence signals co-localized with macroscopic lesions consisting mainly of CD8^+^ T lymphocytes [25,27]. Further supporting the establishment of a persistent infection based on the maintenance of episomal genomes rather than repetitive abortive infections, whole transcriptome analysis of the lymph nodes demonstrated that ORF73 mRNAs could be detected, whereas genes that were known to be essential for viral replication and virion formation could not be detected, such as DNA polymerase or major capsid protein [41]. Thus, although we still do not understand the exact nature of CD8^+^ T-cell dysregulation upon AlHV-1 infection and which viral genes induce lymphoproliferation during WD-MCF, a body of evidence supports that WD-MCF lesions result from the proliferation and infiltration of latently infected CD8^+^ T lymphocytes that resembles a peripheral T cell lymphoma (Figure 6).

## 5. Current Status of Diagnosis and Vaccines

Diagnosis of WD-MCF is commonly based on the identification of clinical signs and lesions, combined with a history of exposure to wildebeests. In addition, detection of specific antibodies can also be useful for epidemiological studies [40]. A widely used multi-species competitive-inhibition ELISA targets the conserved antigenic epitope 15-A of MCF-associated *Macavirus* members [142]. An alternative WC11-based indirect ELISA showed good sensitivity and specificity for detection of MCFV-specific antibodies [143,144]. However, although these ELISA tests have a good specificity to MCFV with low cross-reactivity with other herpesviruses, they cannot distinguish an infection caused by AlHV-1, OvHV-2, or another MCFV due to the shared epitopes in immunodominant viral target proteins. However, one cross-reactive non-neutralizing nuclear epitope has been identified in AlHV-1 and BoHV-4 that could interfere with the serological results [145], although such cross-reactivity did not seem to significantly affect the specificity of the developed serologic tests [144]. Neutralizing antibodies are produced in WD-MCF cases. The antibody titer only increases a few days after MCF clinical signs start, but they do not mediate clinical protection [74,146,147] supposedly because WD-MCF is induced by latently infected proliferating CD8^+^ T lymphocytes in the absence of lytic viral replication. However, neutralizing antibodies could be used to differentiate from sheep-associated MCF. Indeed, neutralizing antibody cross-reactivity exists among viruses within the same subgroup but does not exist between different subgroups [148]. For etiological diagnosis, the availability of genomic sequences of isolated strains has made the use of polymerase chain reaction (PCR) accessible [143,149,150]. In addition, the use of consensus pan-PCR for herpesviruses or MCFV has significantly improved laboratory diagnostics for WD-MCF and has also led to the identification of additional MCF-associated γHVs that had not been yet identified [15,63,151].

**Figure 6 viruses-15-00526-f006:**
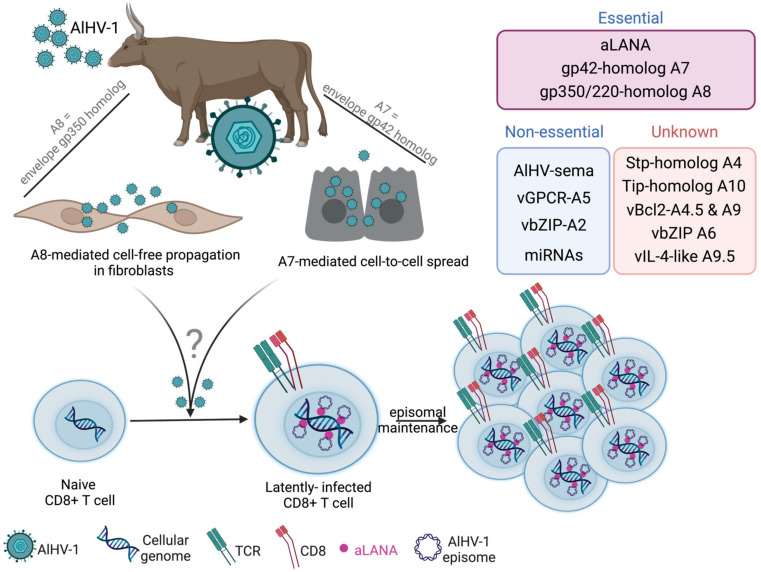
Proposed model for the pathogenesis of WD-MCF, a peripheral T cell lymphoma induced by AlHV-1 infection. AlHV-1 infects WD-MCF-susceptible species via a mucosal route, likely respiratory. Then, host entry is regulated by both A7 and A8 envelope glycoproteins. Following such primary productive infection, CD8^+^ T cells are infected by AlHV-1 though an unknown mechanism. In CD8^+^ T cells, AlHV-1 establishes a latent infection and viral episomal genomes are maintained by ORF73-encoded aLANA in dividing lymphocytes. Subsequently, virus-induced cell proliferation ultimately occurs, leading to typical perivascular infiltrates in most tissues. Candidate specific genes of AlHV-1 are proposed. Created with Biorender.com.

Several vaccine strategies have been developed in cattle or the experimental rabbit model using attenuated or inactivated strains of AlHV-1 and associated or not with incomplete Freund’s adjuvant. Most of the initial vaccine candidates failed to protect against a pathogenic wild-type virus challenge by the intravenous route [74,152,153], despite the presence of significant concentrations of neutralizing antibodies in vaccinated animals. Interestingly, the potential protective antigenic cross-reactivity between BoHV-4 and AlHV-1 has been tested by challenging BoHV-4 pre-infected cattle with pathogenic strain C500 of AlHV-1 [145,154]. Although some protection was observed, the use of BoHV-4 as a candidate vaccine was not explored further. However, more recent vaccine trials have shown more efficacy. Significant protective immunity in 9 out of 10 calves was reported against an intranasal challenge following intramuscular vaccination with a high-passage, attenuated version of strain C500 of AlHV-1 formulated with Freund’s adjuvant [155]. These results were encouraging and comparable results have been obtained using a licensed adjuvant [43]. However, protection was no longer effective nine months after vaccination. Nonetheless, vaccine trials were subsequently led in Tanzania and Kenya using the same high-passage, attenuated strain C500 as a vaccine and reported promising results [42,156,157,158]. Overall, most immunization protocols did induce protective immunity against a challenge by the intranasal route, suggesting the importance of blocking viral infection at the portal of entry. Importantly, infection with a non-pathogenic AlHV-1 strain deleted for ORF73 induced full protection against a challenge with pathogenic strain C500 in rabbits [41]. The use of the AlHV-1 strain deleted for ORF73 is of great interest because the strain is unable to establish a latent infection and is therefore rapidly controlled and cleared. A recent study demonstrated that the AlHV-1 strain deleted for ORF73, where the coding sequence of gB (ORF8) was replaced by OvHV-2 gB, was able to replicate in vitro and provide protection against an intranasal challenge with OvHV-2 [159,160]. Thus, the recombinant ORF73-deleted AlHV-1 strain represents a very promising tool for future vaccine development in the near future.

## 6. Socio-Economic Impact of Wildebeest Derived WD-MCF

Due to its reservoir species, the disease is mainly reported in areas where wildebeests are present. An investigation of the socio-economic impact of WD-MCF demonstrated that the impact of WD-MCF is highly overlooked in the Maasailand, a region of wildebeest migration stretching across southern Kenya and northern Tanzania [161,162,163,164]. Indeed, participatory studies in Maasai populations concluded that WD-MCF could be ranked as the first or fourth disease in term of importance, depending on the proximity of wildebeest populations. In addition to its direct impact due to annual mortality rates (7% to 10%), a series of indirect repercussions are associated with preventive measures of eviction adopted by herders [161,162,163,164]. Because of their knowledge of this disease, herders avoid the rich grasslands of the savannah inhabited by wildebeest herds and graze their cattle at higher altitudes. These more wooded areas are poorer in nutrients compared to the herbal areas in lower lands and the prevalence of bovine theileriosis and contagious bovine pleuropneumonia is higher than in the lowlands [162]. Thus, such changes in grazing areas result in a significant reduction in livestock production rates and additional indirect losses due to other pathogens. In addition, this herding strategy increases the distance travelled by herders and the herd itself [162,163,164,165,166,167]. The direct and indirect consequences of WD-MCF are therefore significant for local populations who largely depend on their livestock for their economic and social well-being. Cleaveland et al. reported that 98% of respondents said they would use a vaccine if it were available [162]. This vaccine would allow them to benefit from a more productive grazing system. However, a tight evaluation of the impact of an effective MCF vaccine should be conducted on the already fragile equilibrium of the wildlife in East and southern Africa where wildebeests migrate every year. For instance, would the implementation of an MCF vaccine in the Maasailand lead to overgrazing of the lowlands by fully protected livestock? A clear evaluation of the predictive implications of vaccination is paramount before implementation. In addition to regions naturally inhabited by wildebeests, cases of WD-MCF have also been reported in zoos across the world when wildebeests are not physically separated from other ruminants [10,28,168]. In zoological gardens and game farms, an effective vaccine would be very valuable to protect MCF-susceptible animals, including sometimes endangered species. Although there is hope for commercial vaccine distribution in the near future in southern Africa [166], the only effective management to avoid WD-MCF for the time being remains physical separation of wildebeests from high-value susceptible species.

## 7. Conclusions and Perspectives

Important discoveries have been made in recent years that have helped us better understand the pathogenesis of WD-MCF. These findings have been obtained essentially since the entire genome of AlHV-1 has been cloned as an infectious and pathogenic BAC [86]. The use of this invaluable tool enabled the generation of recombinant viruses and addressed the role of specific genomic regions and genes in WD-MCF. There is good confidence that AlHV-1 targets CD8^+^ T cells, where it persists as episomal DNA genomes that are maintained in proliferating cells by aLANA. It is expected that CD8^+^ T-cell infection and latency establishment in these cells also occur in wildebeests, the natural host in which AlHV-1 has been adapted to persist without inducing any pathology. An important observation based on published studies investigating the role of specific AlHV-1 genes is that the majority were found to be dispensable for WD-MCF. A potential explanation could be that AlHV-1 has co-evolved with wildebeests, whereas cross-species transmission leading to WD-MCF could be seen as an accident of evolution. The task now is to pursue the effort and identify the main mechanism(s) that could explain lymphoproliferation and disease. A full characterization of infected CD8^+^ T lymphocytes during WD-MCF could potentially provide important information about modified activation pathway(s) upon AlHV-1 infection, which could help not only identify the viral candidate gene(s) involved in triggering such pathway(s) but also unravel how such peripheral T cell lymphoma-like disease is fatal. Among those viral candidate genes, it would be of great interest to investigate the role of the potential oncogenes A4 and A10, as available data on the orthologs SaHV-2 STP and Tip have demonstrated their important role in lymphomagenesis.

Thus, a number of questions remain unanswered: What are the key immunological drivers of protection? What are the viral proteins/elements involved in immune evasion and lymphomagenesis? How, when, and where does AlHV-1 infect CD8^+^ T lymphocytes following primary infection? Future studies combining basic viral biology and pathogenesis, host genomics, and immunology are required to advance our understanding of WD-MCF. We are confident that these findings will provide the basis for developing new prevention and control strategies. As next-generation-based technologies continue to improve, genome-, transcriptome-, epigenome-, metabolome-, proteome-, as well as single-cell-based technologies and the integration of data could provide powerful information about WD-MCF pathogenesis. For example, although “bulk” RNA sequencing has been widely used to identify changes in genes caused by infection and should urgently be applied in the context of WD-MCF, such analyses should also be quickly associated with single-cell RNA-sequencing approaches to elucidate the cellular heterogeneity subject to changes in cellular responses during AlHV-1 infection and WD-MCF development. In addition, using the assay for transposase-accessible chromatin combined with high-throughput sequencing (ATAC-seq) could help determine the regulation of gene expression by investigating chromatin accessibility across the cellular genome of AlHV-1-infected cells. Moreover, epigenetic changes have been reported to regulate RNA fate and have been found to play an important role in γHV infections by favoring propagation or mediating immune evasion [169,170]. Whether reported epigenetic modifications, such as m6A methylation, are involved in AlHV-1 infection remains to be investigated. In-depth investigations of WD-MCF pathogenesis using a system immunology approach will provide researchers the necessary information and tools needed to combat this devastating disease.

## Figures and Tables

**Table 1 viruses-15-00526-t001:** Major members of the subfamily *Gammaherpesvirinae*
^a^.

Genus	Virus Species	Abbreviations	Target Cells	Host Target	Associated Diseases
*Lymphocryptovirus*	Epstein-Barr virus, *Human gammaherpesvirus 4*	EBV, HHV-4	B cells	Human	Infectious mononucleosis, Nasopharyngeal carcinoma, Hodgkin’s and Burkitt’s lymphomas
*Rhadinovirus*	Kaposi’s sarcoma-associated herpesvirus, *Human gammaherpesvirus 8*	KSHV, HHV-8	B cells	Human	Kaposi’s sarcoma, Multicentric Castleman ‘s disease, Primary effusion lymphoma
	Murine gammaherpesvirus 68, *Murid gammaherpesvirus 4*	MHV-68, MuHV-4	B cells/Macrophages	Rodents	B cell lymphomas
	Herpesvirus saimiri, *Saimiriine gammaherpesvirus 2*	HVS, SaHV-2	T cells	Squirrel monkey/non-human primates	T cell lymphomas
*Percavirus*	*Equid gammaherpesvirus 2*	EHV-2	B cells	Horse	Respiratory symptoms
*Macavirus* ^a^	*Alcelaphine gammaherpesvirus 1*	AlHV-1	T cells	Wildebeest/Other ruminants	Malignant Catarrhal Fever
	*Alcelaphine gammaherpesvirus 2*	AlHV-2	Unknow	Hartebeest/Barbary red deer/Bison/Topi	Malignant Catarrhal Fever
	*Alcelaphine gammaherpesvirus 2*-like	AlHV-2-like	Unknown	Barbary red deer	Malignant Catarrhal Fever
	WD-MCFV-oryx	WD-MCFV-oryx	Unknown	Oryx	No disease
	*Alcelaphine gammaherpesvirus 2*	AlHV-2	Unknow	Hartebeest/Barbary red deer/Bison/Topi	Malignant Catarrhal Fever
	*Ovine gammaherpesvirus 2*	OvHV-2	T cells	Sheep/Other ruminants	Malignant Catarrhal Fever
	*Caprine gammaherpesvirus 2*	CpHV-2	Unknow	Goats/White-tailed deer/Sika deer	Malignant Catarrhal Fever
	WD-MCFV–white-tailed deer, *Caprine* *gammaherpesvirus 3*	CpHV-3	Unknow	Goat/White-tailed deer/ Red brocket deer /Reindeer	Malignant Catarrhal Fever
	Ibex-WD-MCFV	Ibex-WD-MCFV	Unknown	Ibex/Bongo/Anoa/Pronghorn	Malignant Catarrhal Fever
	Muskox-WD-MCFV	Muskox-WD-MCFV	Unknown	Muskox	No diseases
	Aoudad-WD-MCFV	Aoudad-WD-MCFV	Unknown	Aoudad	No diseases
	Bovine lymphotropic herpesvirus, *Bovine gammaherpesvirus 6*	BoHV6	B cells	Cattle	No disease
	Porcine lymphotropic herpesvirus 1, *Suid gammaherpesvirus 3*	SuHV3	Unknown	Swine	Post-transplantation lymphoproliferative disorder (PTLD)
	Porcine lymphotropic herpesvirus 2, *Suid gammaherpesvirus 4*	SuHV4	Unknown	Swine	Post-transplantation lymphoproliferative disorder (PTLD)
	Porcine lymphotropic herpesvirus 3, *Suid gammaherpesvirus 5*	SuHV5	Unknown	Swine	Post-transplantation lymphoproliferative disorder (PTLD)

^a^ In the genus *Macavirus*, AlHV-1, AlHV-2, AlHV-2-like, HiHV-1, and oryx-WD-MCFV infect species of the subfamilies *Alcelaphinae*/*Hippotraginae*, while OvHV-2, CpHV-2, CpHV-3, ibex-WD-MCFV, Aoudad-WD-MCF, and Muskox-WD-MCF infect species of the subfamily *Caprinae*. BoHV-6, SuHV-3, SuHV-4, and SuHV-5 belong to the genus *Macavirus* but are not associated with WD-MCF.

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
