# Peer review of "Wildebeest-Derived Malignant Catarrhal Fever: A Bovine Peripheral T Cell Lymphoma Caused by Cross-Species Transmission of Alcelaphine Gammaherpesvirus 1"

_viruses, 2023, doi:10.3390/v15020526_

Round 1

Reviewer 1 Report

This paper reviews the latest research progress of Alhv-1 and MCF, especially explains the mechanism of disease induced by Alhv-1 from the aspects of genome and gene coding protein. The potential role of CD8 + T lymphocytes in the maintenance of viral infection was also explored. It is useful to understand peripheral t-cell lymphoma-like disease induced by ALHV-1 infection. But there are some issues that need to be clarified.

1. Malignant catarrhal fever should be added to the keywords.

2. The subject of this paper is Wildebeest-derived malignant catarrhal fever. So my advice is to introduce MCF first, not viruses. It is recommended that the positions of paragraphs 1 and 2 be swapped, and paragraphs 5 and 6 be swapped.

3. It is recommended that sensitive animals other than cattle, such as deer, antelope, and buffalo, be appropriately supplemented in Figure 2.

4. Is there a serological antigen relationship between Bohv-4 and Alhv-1? Can cross-protective vaccines be considered in vaccine research?

5. In line 614, Conclusions and Perspectives. It is less about conclusions and more about perspectives, and it is recommended that conclusions be refined.

6. Delete future prospects from the subject line 576. It is recommended that you merge future prospects in line 576 and perspectives in line 614.

Author Response

Dear Editor,

We are grateful to the reviewer for the thorough evaluation of our manuscript. We now provide a revised version of our manuscript in which we have addressed all the reviewer comments.

Please find below a point-to-point response to the reviewer 1 comments:

Yours sincerely,

Benjamin Dewals.

--------------------------------

Point-to-point response to the reviewer comments

Reviewer 1.

Comments and Suggestions for Authors

This paper reviews the latest research progress of Alhv-1 and MCF, especially explains the mechanism of disease induced by Alhv-1 from the aspects of genome and gene coding protein. The potential role of CD8 + T lymphocytes in the maintenance of viral infection was also explored. It is useful to understand peripheral t-cell lymphoma-like disease induced by ALHV-1 infection. But there are some issues that need to be clarified.

  1. Malignant catarrhal fever should be added to the keywords.

Author’s response: we have added Malignant catarrhal fever in the keywords

  1. The subject of this paper is Wildebeest-derived malignant catarrhal fever. So my advice is to introduce MCF first, not viruses. It is recommended that the positions of paragraphs 1 and 2 be swapped, and paragraphs 5 and 6 be swapped.

Author’s response: we have swapped sections as suggested, and carefully revised the text for consistence and clarity.

  1. It is recommended that sensitive animals other than cattle, such as deer, antelope, and buffalo, be appropriately supplemented in Figure 2.

Author’s response: we have revised figure 2 (now figure 1) to include the suggested species.

  1. Is there a serological antigen relationship between Bohv-4 and Alhv-1? Can cross-protective vaccines be considered in vaccine research?

Author’s response: we thank the reviewer for the suggestion. Rossiter et al. in 1988 indeed tested BoHV-4 as a candidate vaccine against AlHV-1 with some success as there exists an antigenic cross-reactivity. We have included this information in the revised manuscript.

  1.  In line 614, Conclusions and Perspectives. It is less about conclusions and more about perspectives, and it is recommended that conclusions be refined.

Author’s response: we thank the reviewer for the suggestion. We have refined the conclusion in the revised version of the manuscript.

  1. Delete future prospects from the subject line 576. It is recommended that you merge future prospects in line 576 and perspectives in line 614.

Author’s response: we have deleted the word "prospects" from the title.

Reviewer 2 Report

The manuscript “Wildebeest-derived malignant catarrhal fever: a bovine peripheral T cell lymphoma caused by cross-species transmission of alcelaphine gammaherpesvirus 1 (viruses-2143500)” provides a detailed review about the potential mechanisms of pathogenesis and maintenance of viral genomes in infected CD8+ T lymphocytes. I believe it is a relevant topic and that the manuscript contains important contributions to this field of study. Nevertheless, I do have some major comments:

1. It is worrisome that references are missing throughout the manuscript. It is not possible to understand if the statements and phrases are reflecting the author’s opinions or are based on published scientific studies. References must appear after each mention, at the end of each phrase, not solely at the end of each paragraph.

2. The grammar and text structure require major review from an experienced English speaker that is familiar with medical terms (please refer to the paragraph contained within Lines 172-191). At times, such errors compromise the reader’s experience, and become confusing, such as in (but not limited to):

- Lines 563-566: “These areas are both poor in nutrients compared to herbal areas in lower areas and are the risks of bovine theileriosis and contagious bovine pleuropneumonia transmission are higher, resulting in a significant reduction in livestock production rates and additional indirect losses.”

- Lines 584-586: “In fact, serological detection of AlHV-1 in clinical cases has its limitations as susceptible cattle and rabbits develop detectable neutralizing antibodies only a few days 585 before onset or in surviving hosts [62,134,135].

2. One of the study’s objectives is to “present current hypotheses to explain how AlHV-1 infection induces a peripheral T cell lymphoma-like disease.”; however, the pathologic findings (limited to lines 186-197) are poorly described – both in terms of content and medically corrected terms. Please review.

3. Please provide scale to the macroscopic pictures of organs (Figure 4).

4. Please revise ICTV’s nomenclature rules and correct the names of virus species, families and groups throughout the text. Remember that all viral categories (family, subfamily, genus, species …) should be written in italics. This is extremely important. Please use the updated name of the viral species, e.g., Human gammaherpesvirus 4 instead of “human herpesvirus 4. Regarding alcelaphine gammaherpesvirus 1, please note the difference between the name of the viral species (Alcelaphine gammaherpesvirus 1), and the name of the virus (alcelaphine gammaherpesvirus 1 or wildebeest-associated malignant catarrhal fever virus). https://ictv.global/report/chapter/herpesviridae/herpesviridae/macavirus

5. Please specify along the text if the information provided is related to Wildebeest-derived malignant catarrhal fever or to the malignant catarrhal fever caused by ovine gammaherpesvirus 2. I recommend using an abbreviation, e.g., wd-MCF.

Author Response

Dear Editor,

We are grateful to the reviewer for the thorough evaluation of our manuscript. We now provide a revised version of our manuscript in which we have addressed all the reviewer comments.

Please find below a point-to-point response to the reviewer 2 comments:

Yours sincerely,

Benjamin Dewals.

--------------------------------

Point-to-point response to the reviewer comments

Reviewer 2

The manuscript “Wildebeest-derived malignant catarrhal fever: a bovine peripheral T cell lymphoma caused by cross-species transmission of alcelaphine gammaherpesvirus 1 (viruses-2143500)” provides a detailed review about the potential mechanisms of pathogenesis and maintenance of viral genomes in infected CD8+ T lymphocytes. I believe it is a relevant topic and that the manuscript contains important contributions to this field of study. Nevertheless, I do have some major comments:

  1. It is worrisome that references are missing throughout the manuscript. It is not possible to understand if the statements and phrases are reflecting the author’s opinions or are based on published scientific studies. References must appear after each mention, at the end of each phrase, not solely at the end of each paragraph.

Author’s response: we thank the reviewer for the comment. We have taken care in the revised version to include relevant references for all statements.

  1. The grammar and text structure require major review from an experienced English speaker that is familiar with medical terms (please refer to the paragraph contained within Lines 172-191). At times, such errors compromise the reader’s experience, and become confusing, such as in (but not limited to):
    •  Lines 563-566: “These areas are both poor in nutrients compared to herbal areas in lower areas and are the risks of bovine theileriosis and contagious bovine pleuropneumonia transmission are higher, resulting in a significant reduction in livestock production rates and additional indirect losses.”
    • Lines 584-586: “In fact, serological detection of AlHV-1 in clinical cases has its limitations as susceptible cattle and rabbits develop detectable neutralizing antibodies only a few days 585 before onset or in surviving hosts [62,134,135].

Author’s response: we thank the reviewer for the comment and agree that some parts were not well written and generated ambiguity and confusion. We have now revised the manuscript, and corrected the English.

  1. One of the study’s objectives is to “present current hypotheses to explain how AlHV-1 infection induces a peripheral T cell lymphoma-like disease.”; however, the pathologic findings (limited to lines 186-197) are poorly described – both in terms of content and medically corrected terms. Please review.

Author’s response: We have revised the section to add more infiormation and precision in the description of the pathology induced by AlHV-1 infection in WD-MCF susceptible species.

  1. Please provide scale to the macroscopic pictures of organs (Figure 4).

Author’s response: We have revised the figure accordingly, and also included histological example of the characteristic lesion found in MCF cases.

  1. Please revise ICTV’s nomenclature rules and correct the names of virus species, families and groups throughout the text. Remember that all viral categories (family, subfamily, genus, species …) should be written in italics. This is extremely important. Please use the updated name of the viral species, e.g., Human gammaherpesvirus 4 instead of “human herpesvirus 4. Regarding alcelaphine gammaherpesvirus 1, please note the difference between the name of the viral species (Alcelaphine gammaherpesvirus 1), and the name of the virus (alcelaphine gammaherpesvirus 1 or wildebeest-associated malignant catarrhal fever virus). https://ictv.global/report/chapter/herpesviridae/herpesviridae/macavirus

Author’s response: We have revised the text to comply with the ICTV nomenclature guidelines.

  1. Please specify along the text if the information provided is related to Wildebeest-derived malignant catarrhal fever or to the malignant catarrhal fever caused by ovine gammaherpesvirus 2. I recommend using an abbreviation, e.g., wd-MCF.

Author’s response: We now refer to "WD-MCF" in the revised manuscript for AlHV-1-induced MCF, and specify when MCF refers to another form.